# AUTOREGRESSIVE ENTITY RETRIEVAL

**Nicola De Cao[1,2]\*, Gautier Izacard[2,3,4], Sebastian Riedel[2,5], Fabio Petroni[2]**
[1]University of Amsterdam, [2]Facebook AI Research
[3]ENS, PSL University, [4]Inria, [5]University College London
nicola.decao@gmail.com,{gizacard, sriedel, fabiopetroni}@fb.com

## ABSTRACT

Entities are at the center of how we represent and aggregate knowledge. For instance, Encyclopedias such as Wikipedia are structured by entities (e.g., one per Wikipedia article). The ability to retrieve such entities given a query is fundamental for knowledge-intensive tasks such as entity linking and open-domain question answering. One way to understand current approaches is as classifiers among atomic labels, one for each entity. Their weight vectors are dense entity representations produced by encoding entity meta information such as their descriptions. This approach leads to several shortcomings: (i) context and entity affinity is mainly captured through a vector dot product, potentially missing fine-grained interactions between the two; (ii) a large memory footprint is needed to store dense representations when considering large entity sets; (iii) an appropriately hard set of negative data has to be subsampled at training time. In this work, we propose GENRE, the first system that retrieves entities by generating their unique names, left to right, token-by-token in an autoregressive fashion and conditioned on the context. This enables us to mitigate the aforementioned technical issues since: (i) the autoregressive formulation allows us to directly capture relations between context and entity name, effectively cross encoding both; (ii) the memory footprint is greatly reduced because the parameters of our encoder-decoder architecture scale with vocabulary size, not entity count; (iii) the exact softmax loss can be efficiently computed without the need to subsample negative data. We show the efficacy of the approach, experimenting with more than 20 datasets on entity disambiguation, end-to-end entity linking and document retrieval tasks, achieving new state-of-the-art or very competitive results while using a tiny fraction of the memory footprint of competing systems. Finally, we demonstrate that new entities can be added by simply specifying their unambiguous name. Code and pre-trained models at https://github.com/facebookresearch/GENRE.

## 1 INTRODUCTION

The ability to retrieve the correct entity from large Knowledge Bases (KBs) given a textual input is a fundamental building block for several applications (Ferrucci, 2012; Slawski, 2015; Yang et al., 2018a). Most commercial recommendation systems, for instance, include in their pipelines components to detect and disambiguate entity mentions in open text, in order to isolate relevant concepts from non-meaningful data (Slawski, 2015; Yang et al., 2018a). Another example are chat-bots and question answering systems, that are often equipped with retrieval components to surface specific KB entries (e.g., Wikipedia articles) to find knowledge for sustaining a conversation or answering a question (Ferrucci, 2012; Chen et al., 2017; Lewis et al., 2020b; Roller et al., 2020).

Although there has been extensive previous work on entity retrieval (e.g. Hoffart et al., 2011; Piccinno & Ferragina, 2014; Huang et al., 2015; Le & Titov, 2018; Logeswaran et al., 2019; Broscheit, 2019; Wu et al., 2020, to name just a few) there is a common design choice to most current solutions: entities are associated with a unique atomic label and the retrieval problem can be interpreted as multi-class classification across these labels. The match between input and label is calculated through a bi-encoder (Wu et al., 2020; Karpukhin et al., 2020): a dot product between dense vector encodings of the input and the entity's meta information (such as title and description).

---

\* Work done during internship with Facebook AI Research.

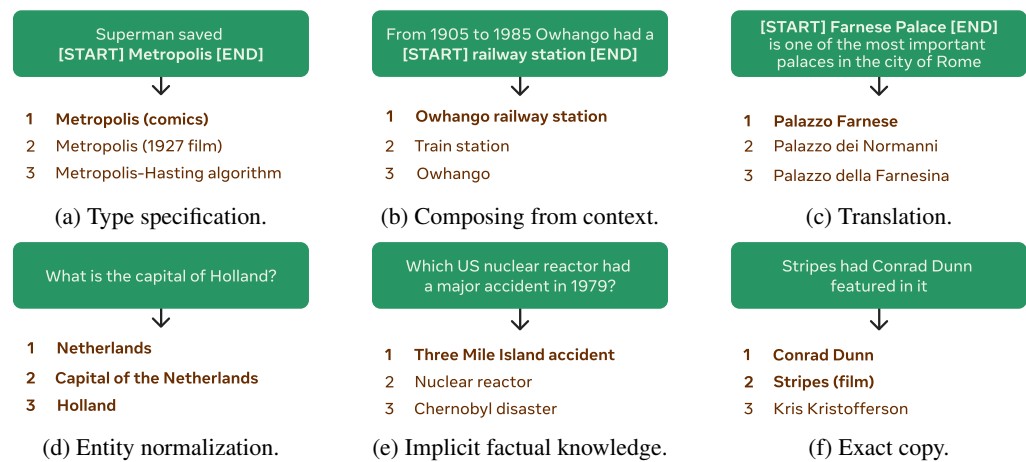

Figure 1: Examples of entities correctly retrieved from GENRE (we show only the top-3 rank). On the *top* three entity disambiguation instances and on the *bottom* three document retrieval instances, two for open-domain question answering and one for fact checking. All of them are cast as sequence-to-sequence problems while inference is done using constrained beam search. Gold entities in **bold**. Sub-captions indicate the type of interaction between the input context and the entity names required.

Critically, this formulation enables sub-linear search using modern maximum-inner-product-search libraries (Johnson et al., 2019) and hence supports retrieving from large entity databases.

Unfortunately, the classifier approach to entity retrieval also has several shortcomings. First, unless a costly cross-encoder is used for re-ranking (Wu et al., 2020), the dot-product can miss fine-grained interactions between input and entity meta information (Humeau et al., 2020). Second, storing dense vectors for the whole KB requires a large memory footprint, especially in real-world scenarios (i.e., ∼24GB to store 1024-dimensional vectors for all of the ∼6M Wikipedia pages), and the size linearly grows with the addition of new entities. Third, computing an exact softmax over all entities is very expensive, hence current solutions need to subsample negative data (Logeswaran et al., 2019; Karpukhin et al., 2020) at training time. Tuning an appropriately hard set of negative instances can be challenging and time-consuming. Finally, existing systems can suffer from a cold-start problem since they cannot represent entities about which they have not yet gathered sufficient information, in the form, for instance, of a textual description or a set of relations with the existing entities.

The treatment of entity identifiers as atomic labels in a classifier ignores the fact that we often have unambiguous, highly structured and compositional entity names. Wikipedia, for instance, associates unique titles to articles,[1] that may be the name of the subject or a description of its topic, as well as potential distinctive information to disambiguate [2] (see Figure 1 for some examples). These entity names often interact with mention contexts in a predictable and regular fashion. For example, often entity names are identical with the mention strings that refer to them (e.g., Fig. 1f). When this is not possible, they might be composed of tokens in the context (e.g., Fig. 1b), include a type specification that can inferred (e.g., Fig. 1a), be the translation of the string mention (e.g., Fig. 1c), require 'normalization' such as referring to the correct alias of a mention (e.g., Fig. 1d), or require factual knowledge that might be stored in the parameters of a model (e.g., Fig. 1e). These observations suggest that inputs could be *translated* into unique entity names, word by word, instead of being classified among a huge set of options.

In this paper, we propose GENRE (for *Generative ENtity REtrieval*), the first entity retriever that exploits a sequence-to-sequence architecture to generate entity names in an autoregressive fashion conditioned on the context. Concretely, GENRE uses a transformer-based architecture, pre-trained with a language modeling objective (i.e., we use BART weights from Lewis et al. (2020a)) and fine-tuned to generate entity names. This architecture has been shown to retain factual knowledge

---

[1]We use *entity name* to refer to the corresponding Wikipedia article title throughout the rest of the paper.

[2]often in the form of a description in parentheses after the name. Wikipedia naming conventions are described in https://en.wikipedia.org/wiki/Wikipedia:Article_titles.

to some extent (Petroni et al., 2019) and language translation skills (Radford et al., 2019) among other things, both desirable properties for an entity retriever. Naturally, the generated output might not always be a valid entity name. To solve this problem, GENRE employs a constrained decoding strategy that forces each generated name to be in a predefined candidate set.

The autoregressive formulation allows us to directly capture the aforementioned relations between context and entity name, effectively cross encoding both. Also, the memory footprint required is orders of magnitude smaller than current systems, since the parameters of a sequence-to-sequence model scale linearly with the vocabulary size, not entity count. Moreover, the exact softmax can be computed efficiently for each output token (i.e., all non-gold tokens are considered negative), thereby eliminating the need for negative data downsampling. Finally, our model never accesses any explicit meta-information about the entity beyond their title, hence new entities can be added by simply appending their unambiguous name to the candidate set (e.g., Fig. 1b refers to an entity added after training).

We empirically evaluate the performance of GENRE on more than 20 datasets, spanning three families of tasks: (i) entity disambiguation, using popular datasets and settings (both in and out-of–domain); (ii) end-to-end entity linking, with the GERBIL benchmarking tool (Röder et al., 2018), by using a novel dynamically markup-constrained decoding strategy; (iii) document retrieval, with the recently proposed KILT benchmark (Petroni et al., 2020b) which spans 5 different sub-tasks. Our models achieve state-of-the-art or very competitive results on nearly all datasets, often with substantial improvement (+13.7 precision points on KILT for retrieval on average). Further, we show that compared with recent models, GENRE requires substantially less memory ($\sim$20 times smaller footprint on average). Finally, we demonstrate that our model can be applied in scenarios where the only entity information available is its name.

We organize the paper as follows: in Section 2 we describe our problem formulation. Then, in Section 3 we present GENRE and eventually in Section 4 we extensively evaluate our method on the aforementioned settings. We will release code and pre-processed data to reproduce our experiments.

## 2 ENTITY RETRIEVAL

We assume to have a collection of entities $\mathcal{E}$ (e.g., Wikipedia articles) where each entity is an entry in a Knowledge Base (KB) such as Wikipedia. We want to approach the following retrieval problem: given a textual input source $x$ (e.g., question), a model has to return the most relevant entities from $\mathcal{E}$ with respect to $x$. We assume that each $e \in \mathcal{E}$ is uniquely assigned to a textual representation (i.e., its name): a sequence of tokens $y$ (e.g., Wikipedia pages are identified by their titles).

A particular instance of this problem is Entity Disambiguation (ED) (see Figure 1 for an example) where an input $x$ is annotated with a mention and a system has to select either its corresponding entity from $\mathcal{E}$, or to predict that there is no corresponding entry in the KB. Another instance is page-level Document Retrieval (DR) where the input $x$ is intended as a query and $\mathcal{E}$ as a collection of documents identified by their unique titles (e.g., Wikipedia articles).

## 3 METHOD

We address the retrieval problem with an sequence-to-sequence model that generates textual entity identifiers (i.e., entity names). Concretely, GENRE ranks each $e \in \mathcal{E}$ by computing a score with an autoregressive formulation: $\text{score}(e|x) = p_\theta(y|x) = \prod_{i=1}^{N} p_\theta(y_i|y_{<i}, x)$ where $y$ is the set of $N$ tokens in the identifier of $e$, and $\theta$ the parameters of the model. We take advantage of fine-tuning the BART (Lewis et al., 2020a) pre-trained language model. We train GENRE using a standard seq2seq objective, i.e., maximizing the output sequence likelihood with teacher forcing (Sutskever et al., 2011; 2014) and regularized with dropout (Srivastava et al., 2014) and label smoothing (Szegedy et al., 2016). Concretely, we use the objective that is typically used for neural machine translation (NMT, Wu et al., 2016), that is maximizing $\log p_\theta(y|x)$ with respect to model's parameters $\theta$ which, due to the factorized formulation, can be calculated exactly. We do not need negative sampling to approximate the loss normalizer.

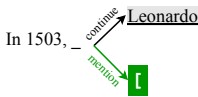 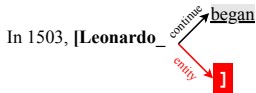 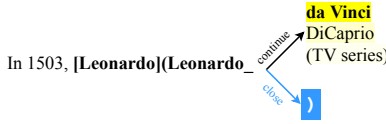

(a) Outside: we can either continue to generate the input or start a new mention.

(b) Inside a mention: we can either continue to generate the input or end the current mention.

(c) Inside an entity link: we can either generate from the entities prefix trie or close if the generated prefix is a valid entity.

Figure 2: Example of dynamically constrained *Markup* decoding for entity linking using "*In 1503, Leonardo began painting the Mona Lisa.*" as input. There are 3 cases: when we are outside a mention/entity (a), inside a mention generation step (b), and inside an entity link generation step (c). The model is supposed to output the input source annotating mentions and pointing them to the respective entities: "*In 1503, [Leonardo](Leonardo da Vinci) began painting the [Mona Lisa](Mona Lisa)*".

## 3.1 INFERENCE WITH CONSTRAINED BEAM SEARCH

Naturally, at test time, we could compute a score for every element in $\mathcal{E}$ and then sort them. Unfortunately, this might be prohibitively expensive when $\mathcal{E}$ is very large (e.g., Wikipedia has ~6M entities). Hence, we exploit Beam Search (BS, Sutskever et al., 2014), an established approximate decoding strategies to efficiently navigate the search space. Instead of explicitly scoring all entities in $\mathcal{E}$, we search for the top-$k$ entities in $\mathcal{E}$ decoding from our model using BS with $k$ beams. Note that using BS implies that the time cost of our retriever does not depend on the size of $\mathcal{E}$, but only on the size of the beams and the average length of entity representations as we do autoregressive generation. The average length of entity representations is tractable (e.g., Wikipedia titles have 6 BPE tokens on average) and we follow standard NMT settings where $k$ is small (e.g., 10).

Since we want to output only entities from $\mathcal{E}$ we cannot use traditional BS while decoding. Indeed, allowing to generate any token from the vocabulary at every decoding step might lead the model to generate output strings that are not valid identifiers. Hence, we resort to Constrained BS, forcing to only decode valid entity identifiers. BS only considers one step ahead during decoding so we can only constrain the generation of a single next token conditioned on the previous ones. Thus, we define our constrain in terms of a prefix tree $\mathcal{T}$ (aka trie) (Cormen et al., 2009) where nodes are annotated with tokens from the vocabulary. For each node $t \in \mathcal{T}$, its children indicate all the allowed continuations from the prefix defined traversing the trie from the root to $t$.

See Figure 9 in Appendix C for an exampled of a trie. When the number of allowed outputs is tractable (e.g., generating a Wikipedia title among ~6M) the trie is relatively small it can be pre-computed and stored into memory (e.g., constraining on Wikipedia titles using the BART tokenizer produces a trie with ~6M leaves, ~17M internal nodes that occupied ~600MB of disk space). We employed the constraints masking the log-probabilities of the invalid tokens and not their logits (i.e., we do not re-normalize the probability over the vocabulary).[3]

## 3.2 AUTOREGRESSIVE END-TO-END ENTITY LINKING

We additionally extend our autoregressive framework to address end-to-end Entity Linking (EL) where, given a document, a system has to both detect entity mentions and link those mentions to their respective KB entities. In this setting, we train the model to predict the source input again but with annotated spans. We use a *Markup* annotation where spans boundaries are flagged with special tokens and accompanied by their corresponding entity identifiers.

Differently from a setting where the output space is relatively small (e.g., a pre-defined set $\mathcal{E}$), the space of annotated outputs is exponentially large. Hence, it is intractable to pre-compute a trie for decoding, and we compute it dynamically instead. In Figure 2 we show an example. At each generation step, the decoder is either generating a mention span, generating a link to a mention, or continuing from the input source. When outside a mention/entity step the decoder has only two options: (i) to continue by copying the next token from the input source, or (ii) to generate the *start of mention* token (i.e., '[') which makes the decoder enter the mention generating phase. While

---

[3]We experimented with both versions and we find masking the log-probability more effective.

generating a mention, the decoder has either to continue with the next token in the input source or to generate the *end of mention* token (i.e., ']') which makes the decoder enter the entity generating phase. Finally, when generating an entity, the decoder employs the entities trie such that it can only output a valid entity identifier as in Constrained Beam Search explained above.

# 4 EXPERIMENTS

We extensively evaluate GENRE on more than 20 datasets across 3 tasks: Entity Disambiguation, end-to-end Entity Linking (EL), and page-level Document Retrieval. We describe the experimental settings in Section 4.1 where we discuss results in Section 4.2. All experiments are in English.

## 4.1 SETTINGS

**Entity Disambiguation (ED)** We reproduce the setting of Le & Titov (2018) using the same candidate sets, *in-domain* and *out-of-domain* datasets, and evaluating using the *InKB* micro-$F_1$. We train GENRE feeding each document where a single mention is flagged with two special start and end tokens and the target output is the textual representation of the corresponding entity. At test time, we decode using constrained beam search with a trie obtained using the provided candidate set (i.e., a subset of $\mathcal{E}$). As large generative models benefit from large amount of data, we first pre-train GENRE on the BLINK data (Wu et al., 2020), i.e., 9M unique triples document-mention-entity from Wikipedia. Then, for the *in-domain* scenario, we fine-tune using the AIDA-CoNLL dataset (Hoffart et al., 2011). For the *out-of-domain* scenario, we evaluate on five test sets: MSNBC, AQUAINT, ACE2004, WNED-CWEB (CWEB) and WNED-WIKI (WIKI) (Gabrilovich et al., 2013; Guo & Barbosa, 2018). More task details and hyperparameters setting are reported in Appendix A.1.

**End-to-End Entity Linking (EL)** For EL, we reproduce the setting of Kolitsas et al. (2018) using the same *in-domain* and *out-of-domain* datasets as well as evaluating the *InKB* micro-$F_1$ on the GERBIL benchmark platform (Röder et al., 2018). Similarly to the ED setting, we first pre-traine our model on all abstract sections from Wikipedia[4] enriched by a string matching heuristic to solve co-references (i.e., if there is a string that matches exactly with another hyperlink we also add it to the dataset as a mention/entity pairs). Then, for the *in-domain* scenario, we fine-tune using the AIDA-CoNLL dataset. We evaluate on seven *out-of-domain* test sets: MSNBC, Derczynski (Der) (Derczynski et al., 2015), KORE 50 (K50) (Hoffart et al., 2012), N3-Reuters-128 (R128), N3-RSS-500 (R500) (Röder et al., 2014), and OKE challenge 2015 and 2016 (OKE15 and OKE16) (Nuzzolese et al., 2015). More task details and hyperparameters setting are reported in Appendix A.2.

**Page-level Document Retrieval (DR)** For this setting, we test GENRE on all the KILT benchmark tasks (Petroni et al., 2020b). Here, whole Wikipedia is used as the candidate set and we evaluate using R-precision (Beitzel et al., 2009). KILT consists of five tasks that use the same Wikipedia dump as a knowledge source: fact checking with FEVER (Thorne et al., 2018); open domain question answering using Natural Questions (Kwiatkowski et al., 2019), HotpotQA (Yang et al., 2018b), TriviaQA (Joshi et al., 2017), ELI5 (Fan et al., 2019); slot filling with T-REx (Elsahar et al., 2018), Zero Shot RE (Levy et al., 2017); entity disambiguation on AIDA CoNLL-YAGO, WNED-WIKI and WNED-CWEB; dialogue with Wizard of Wikipedia (Dinan et al., 2019). We train GENRE on BLINK and all KILT data simultaneously with a single model.[5] More details on the hyperparameter setting are reported in Appendix A.3.

## 4.2 RESULTS

Overall, GENRE achieves very competitive results in all of the three settings being the best performing system on average across all of them. See Appendix C for examples of inputs, ground truth and model predictions for all of the three tasks. In the following, we discuss how GENRE compares to SOTA systems as well as showing some quantitative analysis on its memory footprint, how

---

[4]It is based on the 2019/08/01 Wikipedia dump pre-processed by Petroni et al. (2020b).

[5]Note that not all dataset available in KILT have a training set. Concretely, we train on FEVER, Natural Questions, HotpotQA, TriviaQA, T-REx, Zero Shot RE, AIDA CoNLL-YAGO, and Wizard of Wikipedia.

| | In-domain | | Out-of-domain | | | | |
| Method | AIDA | MSNBC | AQUAINT | ACE2004 | CWEB | WIKI* | Avg. |
|---|---|---|---|---|---|---|---|
| Ganea & Hofmann (2017) | 92.2 | 93.7 | 88.5 | 88.5 | 77.9 | 77.5 | 86.4 |
| Guo & Barbosa (2018) | 89 | 92 | 87 | 88 | 77 | 84.5 | 86.2 |
| Yang et al. (2018a) | **95.9** | 92.6 | 89.9 | 88.5 | **81.8** | 79.2 | 88.0 |
| Shahbazi et al. (2019) | 93.5 | 92.3 | 90.1 | 88.7 | 78.4 | 79.8 | 87.1 |
| Yang et al. (2019) | 93.7 | 93.8 | 88.2 | 90.1 | 75.6 | 78.8 | 86.7 |
| Le & Titov (2019) | 89.6 | 92.2 | **90.7** | 88.1 | 78.2 | 81.7 | 86.8 |
| Fang et al. (2019) | 94.3 | 92.8 | 87.5 | **91.2** | 78.5 | 82.8 | 87.9 |
| BLINK w/o candidate set** | 79.6 | 80.0 | 80.3 | 82.5 | 64.2 | 75.5 | 77.0 |
| **GENRE** | 93.3 | **94.3** | 89.9 | 90.1 | 77.3 | **87.4** | **88.8** |
| Ablations | | | | | | | |
| GENRE only AIDA data | 88.6 | 88.1 | 77.1 | 82.3 | 71.9 | 71.7 | 80.0 |
| GENRE only BLINK data | 89.3 | 93.3 | 90.9 | 91.1 | 76.0 | 87.9 | 88.1 |
| GENRE w/o candidate set | 91.2 | 86.9 | 87.2 | 87.5 | 71.1 | 86.4 | 85.1 |
| GENRE w/o constraints | 86.4 | 80.0 | 81.7 | 82.1 | 66.0 | 81.1 | 79.6 |

Table 1: Micro $F_1$ (InKB) on the in-domain test set and five out-of-domain test sets for the named entity disambiguation task. **Bold** indicates best model and underline indicates second best (not for ablations). *WIKI is usually considered out-of-domain but note that all methods use a part of Wikipedia to train. **results taken from `https://github.com/facebookresearch/BLINK` and normalized to accommodate entities not in KB.

it exploits the structured of the entity name space, and how it behaves on a *cold-start* scenario where new unseen entities are added to the KB (descriptions of those entities are unobserved).

**Comparing GENRE to SOTA systems** In ED the difference in average $F_1$ score between GENRE and the second best performing system is small (i.e., +0.8) however, ED is an established task with more than a decade of research that benchmarked on those datasets. Indeed all systems reported in Table 1 achieved high and similar results even if they were taken from three years back.

The improvements on EL are instead more evident. GENRE is the best in-domain system for AIDA while performing remarkably well also on the out-of-domain setting (e.g., +13 $F_1$ points on Derczynski, and +4.7 on KORE50). Noticeably, in two datasets (OKE15 and OKE16) our model performs poorly. However, these datasets are annotated with coreference (pronouns and common nouns are linked to entities) while our model was not specifically trained for that. Conversely, most of the other systems, have a mention detection component in their pipelines that can be trained or biased to also solve these cases. We considered out of the aim of this work to additional train and evaluate on coreference and we leave it for future work.

On page-level DR, the superiority of GENRE is remarkable. Our model is the best performing system across all 5 KILT tasks and all datasets except on Natural Questions where it is the second best. We achieve +13.7 R-precision points on average with respect to the best performing baseline. In Table 3 we compare GENRE against all methods reported in the public leaderboard: DPR (Karpukhin et al., 2020), DPR+BERT (Devlin et al., 2019), DPR+BART, tf-idf (Leskovec et al., 2014), RAG (Lewis et al., 2020b), and BLINK+flair (Wu et al., 2020; Akbik et al., 2019). No model except ours was trained on the entire KILT dataset at the same time. A RAG model was trained for every single task as well as for DPR+BERT. Note that this gives and advantage to RAG and DPR+BERT to specialize on single tasks where we have only a single model to solve all of them which still performs better. We speculate that multi-task training could have helped since the all tasks share a common objective to retrieve entities. Both DPR and BLINK+flair were not trained specifically on KILT. However, DPR was trained using several QA datasets which include Natural Question and TriviaQA. In Appendix B we report additional results where we do not pre-train or fine-tune our models for both the ED and retrieval setting in Table 1 and 8 respectively. When we train GENRE only in the DPR or BLINK data, our model still outperforms them.

| Method | In-domain | Out-of-domain | | | | | | | Avg. |
|---|---|---|---|---|---|---|---|---|---|
| | AIDA | MSNBC | Der | K50 | R128 | R500 | OKE15* | OKE16* | |
| Hoffart et al. (2011) | 72.8 | 65.1 | 32.6 | 55.4 | 46.4 | **42.4** | 63.1 | 0.0 | 47.2 |
| Steinmetz & Sack (2013) | 42.3 | 30.9 | 26.5 | 46.8 | 18.1 | 20.5 | 46.2 | 46.4 | 34.7 |
| Moro et al. (2014) | 48.5 | 39.7 | 29.8 | 55.9 | 23.0 | 29.1 | 41.9 | 37.7 | 38.2 |
| Kolitsas et al. (2018) | 82.4 | 72.4 | 34.1 | 35.2 | **50.3** | 38.2 | 61.9 | 52.7 | 53.4 |
| Broscheit (2019) | 79.3 | - | - | - | - | - | - | - | |
| Martins et al. (2019) | 81.9 | - | - | - | - | - | - | - | |
| van Hulst et al. (2020)† | 80.5 | 72.4 | 41.1 | 50.7 | 49.9 | 35.0 | **63.1** | **58.3** | 56.4 |
| **GENRE** | **83.7** | **73.7** | **54.1** | **60.7** | 46.7 | 40.3 | 56.1 | 50.0 | **58.2** |

Table 2: Micro $F_1$ (InKB) on the in-domain test set and four out-of-domain test sets for the entity linking task. **Bold** indicates best model and underline indicates second best. *annotated with coreference (note that we do not train/evaluate our model to link pronouns and common nouns). †results from the Wikipedia 2019 setting as opposed to the 2014 setting (older dump and fewer entities).

**Memory Footprint** GENRE is not only performing better than other SOTA models on DR but it has a significant reduction of memory footprint (disk space). In Figure 4 we compare the number of model/index parameter against DPR, RAG, and BLINK. GENRE uses an order of magnitude less parameters (millions instead of billions) to store the entity index because it just has to use a prefix tree of the entity names as opposed to a dense vector for each entity. Concretely, GENRE occupied 14 times less memory than BLINK and 34 times less memory than DPR.

**Exploiting the Structured Name Space** We investigated some properties of GENRE, comparing two variants of our model to BLINK on the ED task (using WNED-KILT validation set): one trained to generate entity names and another to generate numerical identifiers (IDs). All models are trained on the same data and we report results in Figure 5. When there is an exact match between a mention and its entity name, both BLINK and GENRE almost always make an accurate prediction. Different is the case of partial and no match: GENRE performance is much higher suggesting that our model uses the context more effectively, as the autoregressive formulation allows to cross-encode mention context and entity candidates directly capturing fine-grained interactions between the two. Moreover, when we switch to predicting IDs, the performance drops drastically (-20.3 points on average) indicating that it is important that entity names are meaningful, structured and compositional (as they are in Wikipedia) conversely to atomic IDs. Surprisingly, when there is no overlap between a mention-entity pair, performance are still relatively high by using IDs. This suggests that the model is good at memorizing and recalling identifiers even if numeric.

**Ablation study** We here discuss an ablation study on the entity disambiguation task (see Table 1). Due to space limitation, we discuss an ablation study on document retrieval in Appendix B.2. In Table 1, GENRE only AIDA or BLINK data indicates the ablation for which we only train on one of the two datasets (i.e., only fine-tuning). GENRE (full) is also used with constrained decoding (see Section 3) and in combination with a candidate set (as provided by Le & Titov, 2018). GENRE without candidate set denotes ablating the provided (and small) candidate set and therefore using all the entities in the KB (in our case Wikipedia) as candidates. GENRE without constraints indicates ablating constrained decoding which implies no use of the provided candidates set but also unconstrained generation (i.e., the model may generate entity names that are not in the KB). Eventually, using constrained generation and exploiting the candidate sets proved useful. Training only on AIDA data is insufficient to get high $F_1$ (but AIDA is quite small compared to the 9M datapoints of BLINK data).

**Entity frequency** The performance of a model naturally depends on how many times entities appear in the training data. We show the data distribution of the mention-entity frequency in Figure 3. Most of the pairs appears in Wikipedia (10931 / 13354) where 2423 do not (first bin). The average accuracy is 82.5% but noticeable it is higher for mention-entity pairs that are more frequent (right side of the plot). The accuracy for pairs that do not appear in Wikipedia is substantially lower than the average suggesting that those are harder cases (the very end tail of the distribution). The degradation in performance is minimal indicating that our model is good at predicting rare entities.

| Model | Fact Check. FEV | Entity Disambiguation AY2 | WnWi | WnCw | Slot Filling T-REx | zsRE | Open Domain QA NQ | HoPo | TQA | ELI5 | Dial. WoW | Avg. |
|---|---|---|---|---|---|---|---|---|---|---|---|---|
| DPR + BERT | 72.9 | - | - | - | - | 40.1 | **60.7** | 25.0 | 43.4 | - | - | - |
| DPR | 55.3 | 1.8 | 0.3 | 0.5 | 13.3 | 28.9 | 54.3 | 25.0 | 44.5 | 10.7 | 25.5 | 23.6 |
| tf-idf | 50.9 | 3.7 | 0.24 | 2.1 | 44.7 | 60.8 | 28.1 | 34.1 | 46.4 | 13.7 | 49.0 | 30.5 |
| DPR + BART | 55.3 | 75.5 | 45.2 | 46.9 | 13.3 | 28.9 | 54.3 | 25.0 | 44.4 | 10.7 | 25.4 | 38.6 |
| RAG | 61.9 | 72.6 | 48.1 | 47.6 | 28.7 | 53.7 | 59.5 | 30.6 | 48.7 | 11.0 | 57.8 | 47.3 |
| BLINK + flair | 63.7 | 81.5 | 80.2 | 68.8 | 59.6 | 78.8 | 24.5 | 46.1 | 65.6 | 9.3 | 38.2 | 56.0 |
| **GENRE** | **83.6** | **89.9** | **87.4** | **71.2** | **79.4** | **95.8** | 60.3 | **51.3** | **69.2** | **15.8** | **62.9** | **69.7** |

Table 3: R-Precision for page-level retrieval on KILT test data. **Bold** indicates the best model and underline indicates the second best. For our model, we indicated what datasets we used for training.

| Model | Memory | Param. | Index |
|---|---|---|---|
| DPR | 70.9GB | 220M | 15B |
| RAG | 40.4GB | 626M | 15B |
| BLINK | 30.1GB | 680M | 6B |
| **GENRE** | **2.1GB** | **406M** | **17M** |

| Type (support) | BLINK | GENRE | IDs* |
|---|---|---|---|
| Exact match (1543) | 97.8 | 96.6 | 76.0 |
| Partial match (1531) | 70.7 | 86.9 | 63.8 |
| No match (322) | 49.4 | 59.9 | 55.0 |
| Total (3396) | 81.0 | 88.8 | 68.5 |

Table 4: Comparison between retrieval models on memory (disk space) footprint and number of model/index parameters.

Table 5: Different types of matches between mentions and their entity names on the WNED-KILT. *indicates GENRE trained on numerical identifiers.

**Cold-start**   We manually collect 50 Wikipedia articles that were created in 2020[6] to simulate a *cold-start* setting where new entities are added to the KB and the only entity information available is their names. To create ED instances we resort to hyperlinks pointing to those entities in other Wikipedia articles. 19 out of 50 mentions have an exact match with their respective entity names and all of them were correctly classified by GENRE. In combination with the results from Table 5 we can conclude that GENRE has a bias on exactly copying the mention, and this helps on unseen data. GENRE also correctly classified 14/31 of the remaining mentions (45.2%). This demonstrates the ability of our solution to be applied in scenarios where entity metadata is unavailable (apart his name), a setting where, to the best of our knowledge, no existing system is capable to operate.

We additionally test how GENRE performs on unseen mention-entity pairs on WikilinksNED Unseen-Mentions data (Onoe & Durrett, 2020) and we report all results in Table 6 in Appendix B.1. Surprisingly, GENRE performs almost the same for seen and unseen entity pairs (64.4 vs 63.2 accuracy) However, in the Onoe & Durrett (2020) setting we cannot guarantee entity descriptions have not been seen by BART during pre-training (given his training data contains Wikipedia).

## 5   RELATED WORKS

Casting NLP tasks with a structured input or output into sequence-to-sequence problems has been explored for different problems, including semantic parsing (Rongali et al., 2020), semantic role labelling (Daza & Frank, 2018), discourse representation structure parsing (Liu et al., 2018), generation of fluent natural language responses from structured semantic representations (Balakrishnan et al., 2019), generation and parsing of abstract meaning representation (Konstas et al., 2017). In these works a structured representation, a tree or a graph for instance, is linearized into a sequence of symbols compatible with a seq2seq architecture. To the best of our knowledge, we are the first to cast entity retrieval as a sequence-to-sequence problem while decoding with an autoregressive formulation during inference.

Related to our constrained generation mechanism, Daza & Frank (2018); Rongali et al. (2020) use a copying mechanism in order to limit lexical deviations between the input and output strings. In these tasks, as well as for our problem, it is natural to promote a copying mechanism due to the input and the output proximity. A different type of constraint, a structural constraint, is used in Bal-

---

[6]Note that both pre-training and fine-tuning use dumps from 2019.

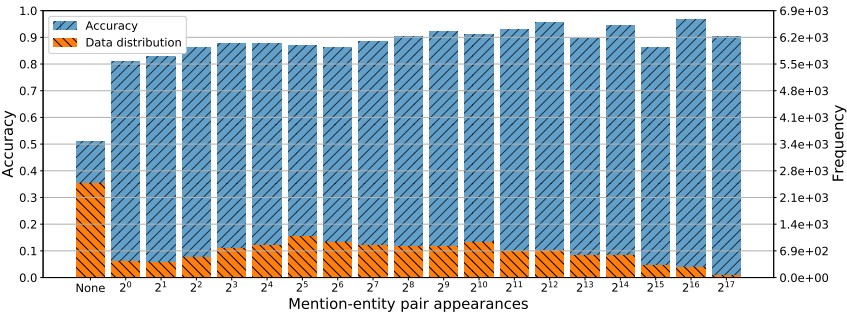

Figure 3: Accuracy per mention-entity pair frequency (in Wikipedia) on the validation sets of all Entity Disambiguation tasks in KILT.

akrishnan et al. (2019) to maintain a valid tree structure. Our constrained beam search encompasses both aspects, a copying mechanism that restrains the vocabulary and a structural constraint to obtain a well-formed annotated output. In addition to these tasks with close input and output, the integration of a mechanism to guide the output of neural networks has been explored in various settings. Lexically constrained decoding has been used to force the inclusion of pre-specified words for machine translation (Hokamp & Liu, 2017; Post & Vilar, 2018), and image captioning (Anderson et al., 2017). To the best of our knowledge, we are the first to exploit constrained generation for entity disambiguation, end-to-end entity linking, and query-based entity retrieval.

Nogueira et al. (2020) propose to use a sequence-to-sequence model to re-rank document. Given a query and a document the model is trained to output the words "true" or "false" depending on whether the document is relevant or not. Differently from our approach for entity retrieval, it requires a limited list of candidates documents, obtained with BM25 for instance, in order to be computationally possible. Massarelli et al. (2019); Petroni et al. (2020a) explore the idea of using an autoregressive language model as neural retriever, by exploiting the implicit knowledge stored in their parameters to generate relevant sentences given a query. While intriguing, such solutions still lag behind retrievers with an explicit knowledge access (e.g., an explicit Wikipedia index). The idea of using a generative model for entity disambiguation was proposed in Petroni et al. (2020b) as they trained both BART and T5 in a seq2seq fashion on all KILT tasks (including ED). We expanded that intuition generalizing on multiple tasks (end-to-end EL and page-level retrieval) as well as introducing constrained decoding for an efficient and effective search.

## 6 CONCLUSIONS

In this work, we propose GENRE, a novel paradigm to addresses entity retrieval: generate entity names autoregressively. Entity names have several properties that might help (even humans) retrieving them, including a compositional structure and a predictable interaction with the context. The autoregressive formulation allows us to directly capture some of these properties, leading to several advantages with respect to current solutions, including an efficient way to cross encode mention context and entity candidates, a much smaller memory footprint, and the ability to compute an exact softmax without the need to subsample negative data. We empirically show that these characteristics, combined with constrained decoding strategies, led to state-of-the-art performance on a plethora of entity retrieval datasets, spanning entity disambiguation, end-to-end entity linking, and page-level document retrieval, while resulting in systems with a remarkably contained memory footprint, a space reduction by a factor of twenty on average. We additionally demonstrate that new entities can be effectively considered in our system by simply appending their unambiguous name to the candidate set.

### ACKNOWLEDGMENTS

Authors thank Patrick Lewis, Aleksandra Piktus, Michael Schlichtkrull, Ivan Titov, Jean Maillard, Edouard Grave, Sergio De Cao, Luisa Quarta for helpful discussions and technical support.

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

# A  EXPERIMENTAL DETAILS

We implemented, trained, and evaluate our model using the `fariseq` library (Ott et al., 2019). We trained GENRE for every task using Adam (Kingma & Ba, 2014) with a learning rate $3 \cdot 10^{-5}$ with a linear warm-up for 500 steps and then liner decay. The objective is sequence-to-sequence categorical cross-entropy loss with 0.1 of label smoothing.

## A.1  NAMED ENTITY DISAMBIGUATION

**Setting**   Given a document $d_j$ (e.g., a sentence) containing a set of entity mentions $m_1, \dots, m_N$, a system either has to assign, to each mention $m_i$, either a KB entity (i.e., $e_i \in \mathcal{E}$), or predicts that there is no corresponding entry in the KB (i.e., $e_i = \text{NIL}$). Moreover, a restricted candidates set $C_i = \{\hat{e}_{i1}, \dots, \hat{e}_{iK}\} \subseteq \mathcal{E} \cup \{\text{NIL}\}$ for each mention $m_i$ is provided.

**Training**   We pre-trained GENRE on BLINK data for 200k steps and then we do model selection on the validation set. Afterward, we fine-tuned on AIDA without resetting the learning rate nor the optimizer statistics for 10k steps and we do model selection on the validation set. Following previous works (Yamada et al., 2016; Ganea & Hofmann, 2017; Le & Titov, 2018), we considered only mentions that have entities in the KB (i.e., Wikipedia). Training was done on 32 GPUs (with 32GB of memory) and it completed in ~24h for a total of ~32 GPU/day.

**Inference**   At test time, we use Constrained Beam Search with 10 beams, and maximum decoding steps of 15. We restrict the input sequence to be at most 384 tokens cutting the left, right, or both parts of the context around a mention. We normalize the log-probabilities by sequence length.

## A.2  ENTITY LINKING

**Setting**   Given a document $d_j$ (e.g., a sentence) a system has to return a set of tuples $\langle m_i, e_i \rangle$ where $m_i$ is a entity mentions (a span contained in $d_j$) and $e_i \in \mathcal{E}$ its corresponding entity in the KB. Following Kolitsas et al. (2018), we considered only mentions that have entities in the KB (i.e., Wikipedia) and we used their candidate sets with the additions of the table computed by Hoffart et al. (2011).

**Training**   We pre-trained GENRE on all abstract sections from Wikipedia[7] enriched by a string matching heuristic to solve co-references (i.e., if there is a string that matches exactly with another hyperlink we also add it to the dataset as a mention/entity pairs) data for 200k steps. Then we do model selection on the validation set. Afterward, we fine-tuned on AIDA resetting the learning rate and the optimizer statistics for 10k steps and we do model selection on the validation set. Again, following previous works (Kolitsas et al., 2018), we considered only mentions that have entities in Wikipedia. Training was done on 64 GPUs (with 32GB of memory) and it completed in ~30h for a total of ~80 GPU/day.

**Inference**   At test time, we use Constrained Beam Search with 6 beams, and a maximum decoding step of 384. When the input sequence is too long, we split the input into multiple chunks of equal size. We normalize the log-probabilities by sequence length.

## A.3  PAGE-LEVEL DOCUMENT RETRIEVAL

**Setting**   Given a query $q$ (e.g., a question) and a collection of documents $\mathcal{D}$ (in KILT are Wikipedia pages), a system has to rank documents in $\mathcal{D}$ based on their relevance to $q$.

**Training**   We trained GENRE on all KILT data simultaneously for 200k steps and we do model selection on the validation set averaging the score across tasks. Training was done on 128 GPUs (with 32GB of memory) and it completed in ~33h for a total of ~176 GPU/day.

---

[7]It is based on the 2019/08/01 Wikipedia dump pre-processed by Petroni et al. (2020b).

**Inference** At test time, we use Constrained Beam Search with 10 beams. For the ED sub-task, we restrict the input sequence to be at most 384 tokens cutting the left, right, or both parts of the context around a mention. We normalize the log-probabilities by sequence length.

# B ADDITIONAL RESULTS

## B.1 NAMED ENTITY DISAMBIGUATION

Table 6 reports evaluation of GENRE on on WikilinksNED Unseen-Mentions data (Onoe & Durrett, 2020). We also report additional results on AIDA from the literature in Table 7.

| | **Seen** | **Unseen** | **Total** |
|---|---|---|---|
| Exact match | 87.48 (751) | 70.36 (2227) | 74.68 (2978) |
| Partial match | 56.39 (1566) | 61.47 (4838) | 60.23 (6404) |
| No match | 41.46 (205) | 45.04 (413) | 43.85 (618) |
| **Total** | 64.43 (2522) | 63.21 (7478) | 63.52 (10k) |

Table 6: Evaluation of GENRE on WikilinksNED Unseen-Mentions data (Onoe & Durrett, 2020). We train on the provided train set and we report accuracy scores (i.e., precision at 1) alongside with the number of supporting datapoints. We report scores splitting the test set in seen and unseen entities as well as in three different matchings between a mention and its gold entity.

| **Methods** | **micro-$F_1$** |
|---|---|
| Guo & Barbosa (2018) | 89 |
| Le & Titov (2019) | 89.6 |
| Yamada et al. (2016) | 91.5 |
| Ganea & Hofmann (2017) | 92.2 |
| Shahbazi et al. (2019) | 93.5 |
| Chen et al. (2020) | 93.5 |
| Yang et al. (2019) | 93.7 |
| Fang et al. (2019) | 94.3 |
| Raiman & Raiman (2018) | 94.9 |
| Mulang' et al. (2020) | 94.9 |
| Yang et al. (2018a) | **95.9** |
| **GENRE** | 93.3 |

Table 7: Additional results on AIDA. We report Micro InKB $F_1$ on test sets.

## B.2 DOCUMENT RETRIEVAL

Table 8 extends Table 3 with additional results (i.e., training GENRE on the numerical identifiers) and an ablation study on the document retrieval task. The purpose of the experiment is to see whether GENRE benefits from the entity names to be meaningful as well as compositional. Numerical IDs do not have that property. In both cases, the model uses its memorizing capabilities but when using IDs the performance is significantly low. Indeed, with IDs the model has no way to generalize nor to use "implicit knowledge" acquired during the unsupervised pre-training. We also ablate the training data. DPR data corresponds to training only on Natural Questions (NQ) and TriviaQA (TQA) as DPR was trained only for QA tasks on those datasets and two extra ones. Note that training on BLINK data corresponds to only training for entity disambiguation. However, every other task share similarities with entity disambiguation and thus the model is also capable to address the other tasks with non-zero performance. For the ablations, underlined cells indicate what are the results on the respective task on which a model was trained for (i.e., GENRE *only BLINK data* was trained only for ED where GENRE *only DPR data* was trained only for QA). The ablation on data suggests that it is beneficial to train on all tasks simultaneously. GENRE without constraints indicates ablating

constrained decoding which implies unconstrained generation (i.e., the model may generate entity names that are not in the KB).

| Model | Fact Check. FEV | Entity Disambiguation | | | Slot Filling | | Open Domain QA | | | | Dial. WoW | Avg. |
|---|---|---|---|---|---|---|---|---|---|---|---|---|
| | | AY2 | WnWi | WnCw | T-REx | zsRE | NQ | HoPo | TQA | ELI5 | | |
| DPR + BERT | 72.9 | - | - | - | - | 40.1 | 60.7 | 25.0 | 43.4 | - | - | - |
| DPR | 55.3 | 1.8 | 0.3 | 0.5 | 13.3 | 28.9 | 54.3 | 25.0 | 44.5 | 10.7 | 25.5 | 23.6 |
| tf-idf | 50.9 | 3.7 | 0.24 | 2.1 | 44.7 | 60.8 | 28.1 | 34.1 | 46.4 | 13.7 | 49.0 | 30.5 |
| DPR + BART | 55.3 | 75.5 | 45.2 | 46.9 | 13.3 | 28.9 | 54.3 | 25.0 | 44.4 | 10.7 | 25.4 | 38.6 |
| RAG | 61.9 | 72.6 | 48.1 | 47.6 | 28.7 | 53.7 | 59.5 | 30.6 | 48.7 | 11.0 | 57.8 | 47.3 |
| BLINK + flair | 63.7 | 81.5 | 80.2 | 68.8 | 59.6 | 78.8 | 24.5 | 46.1 | 65.6 | 9.3 | 38.2 | 56.0 |
| GENRE only BLINK *IDs* | 1.8 | 65.0 | 63.5 | 58.6 | 0.1 | 0.2 | 0.4 | 0.3 | 5.4 | 0.3 | 13.3 | 19.0 |
| GENRE only DPR data | 70.8 | 9.7 | 1.9 | 7.3 | 60.0 | 79.7 | 58.3 | 40.3 | 69.6 | 13.2 | 52.6 | 42.1 |
| GENRE only BLINK data | 28.1 | 82.5 | 88.1 | 69.9 | 44.8 | 66.1 | 15.0 | 16.4 | 25.6 | 6.8 | 38.7 | 43.8 |
| GENRE w/o constraints | 78.9 | 87.2 | 83.2 | 36.5 | 74.4 | 93.6 | 53.3 | 45.2 | 63.7 | 14.3 | 62.7 | 63.0 |
| **GENRE full** | 83.6 | 89.9 | 87.4 | 71.2 | 79.4 | 95.8 | 60.3 | 51.3 | 69.2 | 15.8 | 62.9 | 69.7 |

Table 8: Ablation study on KILT retrieval. We report R-Precision. GENRE only BLINK *IDs* denotes training on BLINK (Wu et al., 2020) data where instead of using the textual entity representation as target we used a numerical ID.

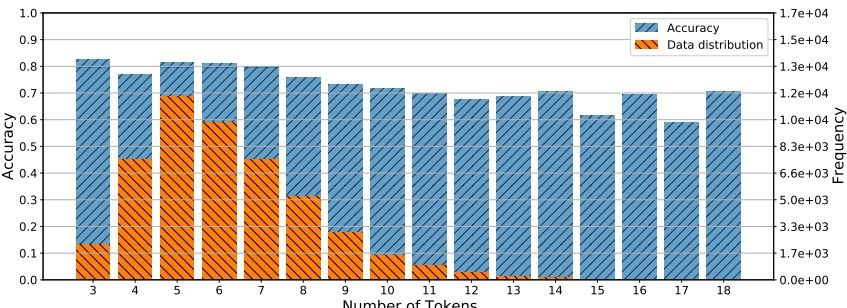

Figure 4: Accuracy per number of BPE tokens of the Wikipedia title to generate on the validation sets of all KILT datasets except ELI5 (as it is fundamentally different from the others). We also show the data distribution of token lengths. Most of the titles have less than 15 BPE tokens while the mode of the distribution is 5. Here GENRE has an average accuracy of 78.6% but it is higher for short titles (e.g., <10) and it is lower for long titles (e.g., ≥10). Degradation in performance does not directly follow the data distribution of the token lengths. Indeed, even if long titles are rare performance is not heavily affected (e.g., for length >15).

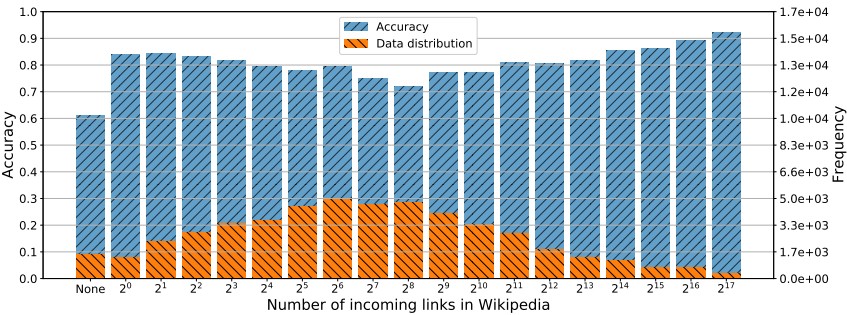

Figure 5: Accuracy per number of incoming links in Wikipedia on the validation sets of all KILT datasets except ELI5 (as it is fundamentally different from the others). We also show the data distribution of the number of incoming links. Intuitively, a page/entity with few incoming links has been observed less than highly connected pages/entities. Indeed, for pages/entities never linked (first bin on the left) the average accuracy is 20% lower than the global average (78.6%). However, for pages/entities linked at least once it is above the global average. This indicates that GENRE seems effective on linking rare entities.

## C  EXAMPLES

```
1   ID : '87d95287−707e−4bd9−9633−ca0c611a4a3a_World_Without_Superma:8'
2   inputs : '[..] When Superman leaves Earth for New Krypton , he appoints , newly freed from
            the Phantom Zone , to take his place as guardian of [START_ENT] Metropolis [END_ENT
            ] . Mon−El assumes the secret identity of Johnathan Kent as a tribute to Clark \'s
            adoptive father , posing as Clark \'s cousin . [..]'
3   gold_output : 'Metropolis (comics)'
4   predicted_outputs : [
5       ('Metropolis_(comics)', −0.09),
6       ('Themyscira_(DC_Comics)', −1.09),
7       ('Metropolis_(disambiguation)', −1.27),
8       ('Superman_(comic_book)', −1.51),
9       ('Superman_(Earth−Two)', −1.52)
10  ]
```

Figure 6: Example of a GENRE prediction for named entity disambiguation on KILT WNED. The input is plain text where a mention is flagged with two special start and end tokens [START_ENT] and [END_ENT]. The output is a ranked list of entity (where we report the log-likelihood as well).

```
1   ID : 'sfq_18245'
2   inputs : "Which Florentine painter
            1535−1607 used the name Bronzino
            after the death of his 'uncle'?"
3   gold_output : 'Bronzino'
4   predicted_outputs : [
5       ('Florence', −0.37),
6       ('Bronzino', −0.62),
7       ('Niccolo_Machiavelli', −0.64),
8       ('Giorgio_de_Chirico', −0.71),
9       ('Vitruvian_Man', −0.73)
10  ]
```

```
1   ID : '4713'
2   inputs : 'Tool has won three Oscars.'
3   gold_output : 'Tool (band)'
4   predicted_outputs : [
5       ('Tool_(band)', −0.08),
6       ('Tool_(disambiguation)', −1.59),
7       ('Machine_Head_(band)', −1.73),
8       ('Language_Arts_(album)', −1.97),
9       ('Machine_Gun_(band)', −2.12)
10  ]
```

(a) TriviaQA (open domain question answering).          (b) FEVER (fact checking).

Figure 7: Example of GENRE predictions for the retrieval task on KILT. The input is a query and the output is a ranked list of Wikipedia article titles (we also report the log-likelihood of the solutions).

```
1   ID: '1106testa_SOCCER'
2   inputs: 'SOCCER − RESULT IN SPANISH FIRST DIVISION. MADRID 1996−08−31 Result of game ↘
        played in the Spanish first division on Saturday: Deportivo Coruna 1 Real Madrid 1.'
3   gold_output: 'SOCCER − RESULT IN [SPANISH](Spain) FIRST DIVISION . [MADRID](Madrid) ↘
        1996−08−31 Result of game played in the [Spanish](Spain) first division on Saturday ↘
        : Deportivo Coruna 1 [Real Madrid](Real Madrid C.F.) 1.'
4   predicted_output: 'SOCCER − RESULT IN [SPANISH](Spain) FIRST DIVISION . [MADRID](Madrid) ↘
        1996−08−31 Result of game played in the [Spanish](Spain) first division on Saturday ↘
        : [Deportivo](Deportivo de La Coruna) Coruna 1 [Real Madrid](Real Madrid C.F.) 1.'
5   gold_spans: [
6       [19, 7, 'Spain'],
7       [44, 6, 'Madrid'],
8       [91, 7, 'Spain'],
9       [147, 11, 'Real_Madrid_C.F.']
10  ]
11  predicted_spans: [
12      [19, 7, 'Spain'],
13      [44, 6, 'Madrid'],
14      [91, 7, 'Spain'],
15      [128, 9, 'Deportivo_de_La_Coruna'],
16      [147, 11, 'Real_Madrid_C.F.']
17  ]
18
19  Micro−precision:    0.80
20  Micro−recall:       1.00
21  Micro−F1:           0.88
```

Figure 8: Example of a GENRE prediction for end-to-end entity linking on AIDA. The input is plain text and the output is a *Markup* string where the links are Wikipedia titles. Spans are in the format $\langle s_i, l_i, t_i \rangle$: *start of the mention*, *length of the mention*, and *title* respectively.

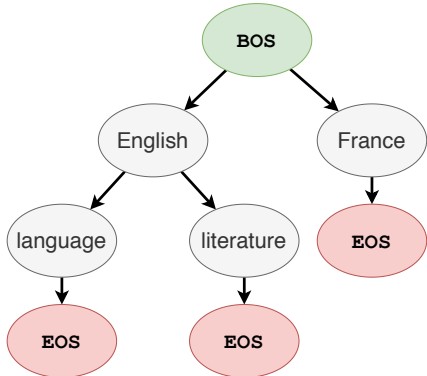

Figure 9: Example of prefix tree (trie) structure where the allowed entities identifiers are 'English language', 'English literature' and 'France'. Note that at the root there is the start-of-sequence token SOS and all leaves are end-of-sequence tokens EOS. Since more that one sequence has the same prefix (i.e., 'English'), this end up being an internal node where branches are the possible continuations.

