# OpenReview forum: "Autoregressive Entity Retrieval"
_ICLR.cc/2021/Conference — ICLR 2021 Spotlight_

### Official Review · AnonReviewer3 · 2020-10-27

**Rating:** 8
**Confidence:** 4

**Review:**

The paper introduces a new method to retrieve entity by auto regressively generating unique entity name as a sequence of word pieces, instead of pinpointing the ID representing an entity. This method stands out in novelty compared to existing various entity retrieval methods, which always assigns a single ID to each entity. Practically, the proposed method has two nice properties: (1) When the entity vocabulary is very large, this approach requires less parameter space and memory compared to other methods (as shown clearly in Table 4) (2) The model can address novel entities, which was unseen during the training. The paper is clearly written and extensively evaluated on three relevant tasks, entity disambiguation, entity linking, and entity retrieval.

I have one big concern with the current format of the presentation. In the current draft, It is not very clear whether the strong gain is coming from large scale pretr aining or the proposed method itself.  To this end, the ablations shown in Table 7 and Table 8 should be reported in the main paper, with clear explanations. As we all know, the model architecture (which is the focus of this paper) cannot be properly evaluated when the training set up is different (i.e., how much pretraining has been done, on what dataset?).  Could you elaborate on this? In appendix, there's only result tables without in-line explanations.

The experiments on cold start, as well as table 5 which shows performances on entities divided by name match is pretty cool!

If space allows, adding some more analysis on what types of entities do this model do better compared to other methods would be interesting (would lengthier names easier or harder? would it do better on popular entities or more long tail ones? are these systems complementary to existing methods or mostly succeed and fail on the same set of examples?)

---

> ### Author Response · Authors · 2020-11-17
> **Reply to R3**
>
> We thank R3 for the useful comments.
>
> **Analysis of ablations**
>
> The focus of the paper is on a new autoregressive way of predicting entity identifiers based on their unique names. We use a standard transformer architecture as several of the baselines we considered (e.g.,  Wu et al., 2019; Karpukhin et al., 2020). The key difference is that current solutions estimate the match between input and entity label though a bi-encoder --- a dot product between dense vector encodings of the input and the entity’s meta information --- that can miss fine-grained interactions between input and entity meta information. Our autoregressive formulation allows to cross-encode mention context and entity candidates directly capturing fine-grained interactions between the two.
>
> Thanks for pointing our that our baselines and setting where not detailed enough, we added more details on the ablation experiments and commenting result tables. We also provide details on which datasets we pre-train our models, for how many steps and other training hyper-parameters (Appendix A). In the next revision we will use extra space in the main paper to report these details.
>
> **Analysis of “tail entities”**
>
> Thanks for suggesting such an analysis, we believe it strengthened our paper. In our last revision, we added 3 new plots (Figures 3, 4 and 5). It turns out that there is a drop in performance for unseen entities but not dramatic. Besides, it seems that GENRE is robust at identifying entities that appear even just once in Wikipedia.  Please refer to the new draft and the general response for further details.
>
> **References**
>
> Wu, L., Petroni, F., Josifoski, M., Riedel, S., & Zettlemoyer, L. ; Scalable Zero-shot Entity Linking with Dense Entity Retrieval. In Proceedings of the 2020 Conference on Empirical Methods in Natural Language Processing (EMNLP). 2020.
>
> Karpukhin, V., Oğuz, B., Min, S., Wu, L., Edunov, S., Chen, D., & Yih, W. T. ; Dense Passage Retrieval for Open-Domain Question Answering. In Proceedings of the 2020 Conference on Empirical Methods in Natural Language Processing (EMNLP). 2020.

---

### Official Review · AnonReviewer2 · 2020-10-28
**Novel idea on entity retrieval/linking with well executed experiments**

**Rating:** 8
**Confidence:** 4

**Review:**

### Summary

This paper proposes to tackle the entity linking task using a sequence-to-sequence neural model, trained by producing unique entity names, in autoregressive fashion. The paper makes a case that this approach can scale better with larger entity vocabularies than previous methods with dedicated entity representations both in terms of memory as well as computation costs. The model is studied under a number of tasks including entity disambiguation, entity linking and document retrieval for question answering.

### Strong and weak points

Strong points:
- The work is well motivated: I believe there are, or will be, many systems where retrieving information about billions of entities is useful. Making simpler and more efficient models to deal with larger entity vocabularies is important.
- Doing seq2seq entity linking is really a novel idea and is surprising that it works so well for several of the datasets presented.
- The idea of constrained decoding using a Trie is neat and makes intuitive sense.
- The empirical evaluation in the paper is quite exhaustive, in terms of number of datasets.

Weak points:
- There is no discussion or analysis on the performance of this new model on “tail entities” (entities that have few examples in the training set). A believe such a discussion would be interesting for two reasons: (1) if one wishes to use this type of model for much larger entity vocabularies, it is likely that a larger fraction of entities will have low number of examples, and (2) one effect of contrastive learning (e.g., negative sampling) has on systems with explicit entity representations is some implicit training of _all_ entities, which is lacking in the described autoregressive proposal.
Finally on this topic of tail entities: the “IDs” experiment, shown in Table 5, indicates that when entity mention and decode target are not related, performance suffers. Similarly, for “tail entities” that are ambiguous with another “popular entity”, the model may be biased with the popular entity (results from “Cold Start” may support this hypothesis). Hard to say without some analysis.

- While the constrained decoding is really interesting, it was not clear from the paper how crucial this was for good performance. Is this something absolutely critical for performance overall, or does it provide a modest performance improvement? While it may be clear to the authors, it would be highly informative to explicitly describe the performance impact on unconstrained beam decoding.

### Recommendation

Overall, my recommendation is to accept this paper to ICLR. I believe the problem of representing entities in natural language systems is of high interest to the ICLR and NLP communities. The ideas in this paper are novel, the paper is well written and the empirical evaluation is well executed.

### Questions for authors

- See comment above regarding analysis of “tail entities”. Could any further insights on this topic be added to the paper?
- See comment above about ablating the constrained beam decoding. This would not need to be a long or complex analysis. Simply 1 or 2 data points for the reader to understand the magnitude of the importance of this decoding.

---

> ### Author Response · Authors · 2020-11-17
> **Reply to R2**
>
> We thank R2 for the useful comments.
>
> **Analysis of “tail entities”**
>
> Thanks for suggesting such an analysis, we believe it strengthened our paper. In our last revision, we added 3 new plots (Figures 3, 4 and 5). It turns out that there is a drop in performance for unseen entities but not dramatic. Besides, it seems that GENRE is robust at identifying entities that appear even just once in Wikipedia.  Please refer to the new draft and the general response for further details.
>
> **Ablating the constrained beam decoding**
>
> Constrained beam decoding is a central part of our system as without it the model may generate entity names that cannot be mapped to a specific entity (e.g., if the model predicts “Michael Jordan (basketball player)” it will not be mapped to the Michael Jordan entity in Wikipedia as the label for that is simply “Michael Jordan”: https://en.wikipedia.org/wiki/Michael_Jordan).  We thank the reviewer for suggesting investigating its importance. We added two new ablations experiments for entity disambiguation and page-level retrieval (on all 6 dataset for ED and all 11 datasets for retrieval — see “GENRE w/o constrains” in Table 7 and 8 respectively). Without  constrained decoding we have an average drop of ~9 F1 points for entity linking (88.8 -> 79.6) and ~7 R-precision points for retrieval (69.7 -> 63.0). Due to space constrains we kept ablations in the Appendix. For future versions (e.g., camera ready) we will move them in the main paper accompanied with comments in the main text.

---

### Official Review · AnonReviewer1 · 2020-10-28
**Review #1: Great paper with convincing results.**

**Rating:** 8
**Confidence:** 4

**Review:**

The paper proposed to use autoregressive approach to solve entity-based problems. They proposed a uniform framework and showed that their model achieved the state of the art performance on 3 different types of tasks (~20 datasets). The GENRE model also significantly reduced the memory usage compared to previous models that stored a big memory table. It's also capable of linking novel entities at inference time. This paper is clearly written. The experiment results are convincing.

One limitation of this paper is that they required the vocabulary of the entities to be the ones that have a Wikipedia page. And that their model relied on copying the surface form of entities (as suggested in the paper). From the experiments, the "copying" approach worked very well on Wikipedia entities, that are often common entities. How can you improve the model to link rare entities not in the Wikipedia pages?

Another concern is the efficiency at inference time. Compared to the models with a large entity memory whose retrieval is performed with Maximum inner product search, how is the efficiency of your decoding strategy?

---

> ### Author Response · Authors · 2020-11-17
> **Reply to R1**
>
> We thank R1 for the useful comment!
>
> **How can you improve the model to link rare entities not in the Wikipedia pages?**
>
> An assumption of our method (and generally all entity linking/ retrieval systems) is that entities belongs to a  Knowledge Base (e.g., entity linking has to point to an entity in the KB). Thus, we require as part of our system to have a set of entity names one for each entity in the KB. However, we do not required the vocabulary of the entities to be the ones that have a Wikipedia page. One can define a different set of entities and it has been the case in some of our experiments (e.g., for end to end entity linking in Table 2, we used the candidate sets provided by Kolitas et al., 2018 which are not all Wikipedia entities). We will clarify this aspect in the paper.
>
> Thanks for suggesting a rare or “tail” entities analysis, we believe it strengthened our paper. In our last revision, we added 3 new plots (Figures 3, 4 and 5). It turns out that there is a drop in performance for unseen entities but not dramatic. Besides, it seems that GENRE is robust at identifying entities that appear even just once in Wikipedia.  Please refer to the new draft and the general response for further details.
>
> **How is the efficiency of your decoding strategy?**
>
> Although time efficiency was not an objective of our work, we agree that it is an interesting thing to know/ add. We measured time efficiency (sentences/sec) on the Entity Disambiguation task against BLINK (Wu et al., 2019). The experiment was done on CPU (since the index occupies ~24GB  we could not run the dot product search on GPU) with top-k=100 for BLINK and beam-size=5 for our method. BLINK has a bi-encoder that retrieves the top-k entires with maximum inner product search and then a cross-encoder that re-ranks the entities with a more complex function. We compare to both versions on the AIDA dataset (note that BLINK bi-encoder only is fast but it has lower performance than BLINK full — see Wu et al., 2019):
>
> | Model |examples/sec |
> |---|---|
> | BLINK (bi-encoder only) | 6.6 |
> | BLINK (full) | 0.6 |
> | GENRE (ours) | 2.9 |
>
> GENRE is ~2x slower than BLINK (bi-encoder only) but ~5x faster than the full BLINK system. We will add these results to the main paper and perhaps with additional comparisons. In future work we might try some techniques from neural machine translation to make the decoder faster.
>
> **References**
>
> Wu, L., Petroni, F., Josifoski, M., Riedel, S., & Zettlemoyer, L. ; Scalable Zero-shot Entity Linking with Dense Entity Retrieval. In Proceedings of the 2020 Conference on Empirical Methods in Natural Language Processing (EMNLP). 2020.

---

### Official Review · AnonReviewer4 · 2020-10-29
**The paper proposes a brand new approach for entity retrieval, which leverages an encoder-decoder architecture to generate the target entity directly. Intensive experiments on entity disambiguation, entity linking, and document retrieval tasks prove the effectiveness of the approach.**

**Rating:** 7
**Confidence:** 5

**Review:**

The paper proposes a brand new approach for entity retrieval, which leverages an encoder-decoder architecture to generate the target entity directly. To tackle the problem of invalid generation, a trie-constrained beam search is used for decoding. The author performs intensive experiments on entity disambiguation, entity linking, and document retrieval tasks and achieves new SOTA or competitive results on over 20 datasets. Although the paper does not come up with new architecture or elaborately designed neural components, I believe this paper is worth reading for the community, including how this paper redefines the problem.

Strengths:
1. A brand new perspective on entity retrieval. Clever use of text generation methods.
2. Compared to previous methods, GENRE is memory saving.
3. This paper provides comprehensive experiments on 3 important tasks, over 20 datasets, showing the robustness of this method.

Weaknesses:
1. Lack of clarity at many places:
a) It's clearer to demonstrate how you reformulate the input and output for the 3 tasks, as they seem very different compared to previous methods
b) It's better to add some description about the baseline systems
c) I'm especially interested in the decoding details on end-to-end entity linking,  it's better to use a formular to demonstrate how you compute the log-probabilities.

Questions:
1. In the entity disambiguation task, it seems that for each inference step, GENRE only generates one entity. Is it possible to generate all entities simultaneously as what GENRE does in end-to-end entity linking?
2. From table 5, an impressive result is that it seems that the pre-trained model is good at memorizing, even if the identifier is numeric. But from table 3, some of the baseline systems also benefit from pretrained models, while achieve much lower results. What do you think causes this difference?

---

> ### Author Response · Authors · 2020-11-17
> **Reply to R4**
>
> We thank R4 for the useful comments.
>
> **Lack of clarity at many places**
>
> a) Thanks for pointing our that the input formulation was not clear. We actually do show how we reformulate the input and output for the 3 tasks in the Appendix (see Figure 3, 4, and 5 of our first submission - on the revision are Figure 6, 7, and 8). We will further clarify this aspect in the final version.
>
> b) We will include brief descriptions of baseline models in further versions (we haven’t included them here due to space constraints as we compare to ~20 other systems from the literature across the 3 tasks).
>
> c) The log-probabilities for all of the 3 tasks are computed according to the equation at the beginning of Section 3: $score(e|x) = p(y|x) = \prod_{i=1}^N p(y_i|y_{<i},x)$. When we do entity disambiguation x is the source document with an annotated mention and y is the entity name. For end-to-end entity linking x is the source document and y is the markup annotated source document (as defined in Section 3.2). The log-probability of an annotation is computed using the equation above token-by-token.
>
> **Is it possible to generate all entities simultaneously as what GENRE does in end-to-end entity linking?**
>
> In the entity disambiguation task GENRE only generates one entity at a time as we process the sequence left-to-right. However, we batch the generation to speed up the computation. Some documents have too many entities to allow inference for all of them simultaneously. Also for end-to-end entity linking GENRE only generates one entity at a time since it is an autoregressive model. The model predicts the markup annotated source token by token so it cannot generate all entities simultaneously. Also for this task we batched documents to speed up computation.
>
> In case you meant whether we have a “global” model (in which predictions depend on each other but where mentions are given) then do not have a “global” model as predictions are only conditioned on previously generated tokens (for the end-to-end entity linking task). We could extend the entity disambiguation model to be global condition the generation of the next token dependent on all the mention/entity pairs. Although this was not a goal of this work.
>
> **What causes the difference between our method and baseline systems as all should benefit from pre-training?**
>
> We consider several baselines that use pre-trained language models (e.g., all those in Table 3 except tf-idf). As we argue in the introduction, we believe that the difference in performance is due to the novel autoregressive way of formulating the output in GENRE. In fact, all current solution estimate the match between input and entity label though a bi-encoder --- a dot product between dense vector encodings of the input and the entity’s meta information --- that can miss fine-grained interactions between input and entity meta information. Our autoregressive formulation allows to cross-encode mention context and entity candidates directly capturing fine-grained interactions between the two.

---

### Author Response · Authors · 2020-11-17
**General Author Response**

As also pointed out by all reviews, an analysis for rare or “tail” entities would add value to the paper. We thank all reviewers for suggesting this. We added 3 new plots (Figures 3, 4 and 5) for analysing such “tail entities”

**Analysis of “tail entities”**

Figure 3 shows the accuracy per mention-entity pair frequency in Wikipedia for the KILT entity disambiguation tasks. The accuracy for pairs that do not appear in Wikipedia is substantially lower than the average (~51% vs ~82%)  suggesting that those are harder cases (the very end tail of the distribution). However, for pairs that are present at least one time in Wikipedia, the accuracy is ~80% meaning that rare (but not absent) mention-entity pairs are predicted well by our model. Note that when a pair appears at least one time in Wikipedia it does not necessary implies we have trained with that pair. But it means that during masked language model pre-training the model has seen that mention at least one time (but not the corresponding entity as we fine-tune for this). Also note that other systems (e.g., BLINK and DPR -- Wu et al., 2019; Karpukhin et al., 2020) have to see all entity descriptions after training to make predictions where we do not.

Figure 4 shows the accuracy per number of BPE tokens in the entity. We also show the data distribution of token lengths. Here GENRE has an average accuracy of 78.6% but it is higher for short titles (e.g., <10) and it is lower for long titles (e.g., >=10). Degradation in performance does not directly follow the data distribution of the token lengths. Indeed, even if long titles are rare performance is not heavily affected (e.g., for length >15).

Figure 5 shows the accuracy per number of incoming links in Wikipedia (i.e., how many times they are referred from other pages) for all KILT tasks. Intuitively, a page/entity with few incoming links has been observed less than highly connected pages/entities. Indeed, for pages/entities never linked the average accuracy is ~20% lower than the global average (78.6%). However, for pages/entities linked at least once it is above the global average.

These three analysis indicates that GENRE seems effective on linking rare (but not absent) entities.

---

### Decision · Program_Chairs · 2021-01-07
**Final Decision**

**Decision:**

Accept (Spotlight)

**Comment:**

This is a novel, simple, and experimentally well-supported new idea for entity linking.  The key insight is to perform entity linking by producing meaningful entity names with seq2seq approaches, and the big surprise is how well this works experimentally (at least for wikipedia-style entities).  Very nice paper!